# SCALABLE SELF-IMPROVING DIFFUSION MODELS

## ABSTRACT

Scalable self-improving capability is a desirable property of generative models, as it enables performance improvement with additional computational resources rather than more training data. Existing approaches typically rely on external reward signals or unstable generation-verification framework to build the improving process. In this paper, we discard the generation-verification gap framework and propose a brand-new self-improving framework for diffusion models (DMs), Scalable Self-Improving Correction (SSI-Corr). Instead, SSI-Corr trains a corrector that directly aligns the sample generation process with the target distribution. Our method is supported by theoretical analysis and scales effectively with available computational resources. In the experiment, we demonstrate that SSI-Corr improves FID scores by 27% and 14.3 % on a pre-trained unconditional CIFAR10 DDPM with ancestral and DDIM samplers respectively.

## 1 INTRODUCTION

Nowadays, self-improving scalability has become an emerging topic (Huang et al., 2025; 2023; Alemohammad et al., 2024b; Yuan et al., 2024), especially as all existing data has already been consumed (Villalobos et al., 2024). Therefore, two questions have become central in the community: (1). can a model self-improve without external data or models, and (2). can its performance scale with increasing computational budgets? However, current works on training and testing scalability either rely on additional reward models for guidance (Zhang et al., 2025; Ma et al., 2025; Singhal et al., 2025). Moreover, existing works on self-improvement of diffusion models (DMs) (Yuan et al., 2024; Alemohammad et al., 2024b) rely on generation-verification gap (Huang et al., 2025) or on self-consuming degradation (Alemohammad et al., 2024a) without theoretical guarantee that the training iteration can reduce the mismatch of generative and ground-truth distributions.

We ask the following question:

*can we design a self-improving DM with train- and test-time scalability?*

Unlike the popular generation-verification gap framework (Huang et al., 2025; Yuan et al., 2024) that requires unstable GAN-like Goodfellow et al. (2014); Radford et al. (2016) bi-level optimization, we approach this problem by revisiting the error decomposition of the diffusion processes (Chen et al., 2023b;a): (1). initial distribution mismatch; (2). score estimation error; (3). discretization error. We propose to mitigate these errors by training a flow matching model (Lipman et al., 2023; Liu et al., 2022b) to interpolate: *the source distribution*: $\hat{p}_t$, the state distribution of the diffusion sampler at time $t$, and *the target distribution*: $p_t$, the true marginal distribution at the same noise level. As a result, the training process of our method only **involves a simple least-squares regression without unstable bi-level optimization** Yuan et al. (2024). This corrector functions as a subroutine in inference to push the state distribution closer to the true marginal distribution at time $t$. Applying the corrector at different noise levels enables **training- and testing- scalability**:

- **Accumulated discretization and score estimation error bring training scalability**: The discretization and score estimation error accumulate through the sampling process and peak near the end. Applying the corrector at the end has the largest potential to mitigate all three errors jointly and improve the sample quality. However, both the source and the target distribution of the flow matching corrector become less smooth and multi-modal. Learning such a corrector can be challenging.

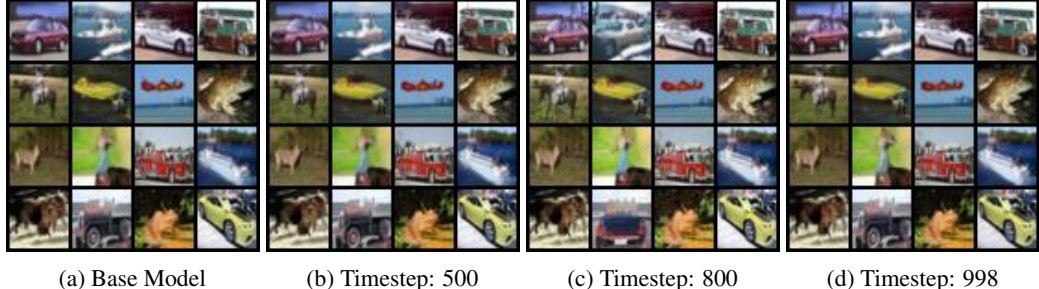

|  (a) Base Model | (b) Timestep: 500 | (c) Timestep: 800 | (d) Timestep: 998 |

Figure 1: CIFAR10 samples generated by DDPM and our method with various correction timesteps.

- **Unlimited synthetic error samples brings self-improving capability**: the error samples to the corrector can be sampled by running the diffusion model sampling process. Without the computation bottleneck, one can generate samples from the corrector source distribution of arbitrary sizes. This sampling procedure scales up the training dataset and enables the training scalability;

- **The non-smoothness of the marginal distribution brings test-time scalability**: The generative and ground-truth marginal distributions become less smooth and multi-modal as the timestep goes up, leading to a complex corrector that requires more test-time compute to sample accurately.

With such design, a user can flexibly allocate training and testing budgets to achieve the desired performance. For example, if a user has a limited training and testing budget, they can train the corrector at the earlier timestep to achieve the best performance with highest computation-performance efficiency. As the feasible computational resource increases, the user can further train a corrector at a later timestep to further improve performance flexibly.

We summarize our contributions as follows:

- We propose SSI-Corr, a **new self-improving framework for diffusion models beyond the generation-verification gap framework** Huang et al. (2025; 2023) by error correction analysis with theoretical guarantee;

- Our method enables **training and testing scalability** for arbitrary compute budget by choosing timestep properly; correction on later timestep requires more synthetic samples and training budget but brings larger performance gain while earlier timesteps brings best computational efficiency for limited budget;

- We conduct a comprehensive evaluation on our method to validate the training and testing scalability, showing our method achieves **27% improvement in FID score** from 6.872 to 4.963 with double test-time compute budget and 50K synthetic samples for DDPM pre-trained on CIFAR10 Krizhevsky (2009) (See Figure 2).

## 2 RELATED WORK

### 2.1 DIFFUSION MODEL AND FLOW MATCHING

DMs learn to reverse a tractable forward corruption process (Sohl-Dickstein et al., 2015; Song et al., 2021b), which does not require special architecture design like flow-based models (Dinh et al., 2017; Kingma & Dhariwal, 2018; Rezende & Mohamed, 2015). Recent works mainly focus on sampling acceleration like Zhao et al. (2023); Lu et al. (2022); Song et al. (2021a); Liu et al. (2022a), and distillation (Song et al., 2023; Song & Dhariwal, 2024; Frans et al., 2025). Flow matching model (Lipman et al., 2023; Liu et al., 2022b; Pooladian et al., 2023; Tong et al., 2024), on the other hand, define a ODE that interpolates the source and target distributions.

| Total Steps | SSI-Corr. (Ours) | PC w/ Bias Corr. | PC w/o Bias Corr. |
|---|---|---|---|
| 1000 | 6.872 | 6.742 | 6.872 |
| 1010 | 7.111 | – | – |
| 2000 | 4.963 | 6.725 | 6.721 |
| 3000 | – | 6.701 | 6.706 |
| 6000 | – | 6.674 | 6.702 |
| 11000 | – | 6.824 | 6.801 |

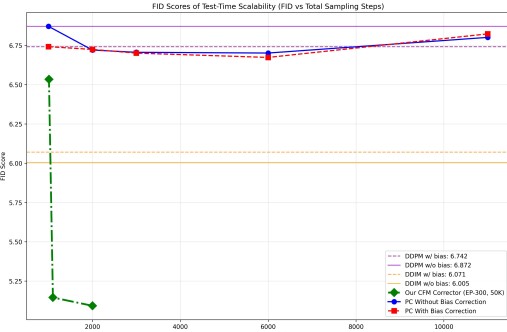

Figure 2: FID (↓) v.s. Total Sampling Steps (sum of predictor and corrector steps). **PC** means the predictor-corrector framework Song et al. (2021b). **Bias Corr.** indicates if enabling dataset mean for tackling the initial bias mismatch.

## 2.2 SELF-IMPROVING GENERATIVE MODELS

Self-improving (Huang et al., 2025; 2023; Tian et al., 2024; Qu et al., 2024) capability is a desired feature of AI models, leveraging the generation-verification gap to improve models with only growing compute and no external data required. Existing works mainly focus on designing more effective reflection prompts (Tian et al., 2024; Qu et al., 2024) for better improvement. However, such methodology does not apply to diffusion models due to the non-readable visual prompt. Existing methods on self-improving diffusion models focus on the generation-verification gap framework Huang et al. (2025) or the self-consuming degradation method Alemohammad et al. (2024b).

## 2.3 SCALING LAWS FOR GENERATIVE MODELS

The training-time scaling laws for neural language models (Kaplan et al., 2020; Hoffmann et al., 2022) found that the model capabilities scale with the growing compute and dataset size. Meanwhile, Liang et al. (2024); Yin et al. (2024) investigated scaling laws for video diffusion models. However, such scaling laws require external training data, which is unrealistic due to data limitations (Villalobos et al., 2024).

As for the test-time scaling, Muennighoff et al. (2025); Snell et al. (2025) has discovered that scaling the test-time computation is more effective than scaling the training-time computation for large language models (LLMs) (Kaplan et al., 2020). Existing test-time scaling on diffusion (Ma et al., 2025; Zhang et al., 2025; Singhal et al., 2025) still relies on additional reward models as guidance, unlike the scalable Best-of-N strategy for LLMs (Jinnai et al., 2025; Ichihara et al., 2025).

## 2.4 ERROR CORRECTION IN GENERATIVE MODELS

Zhang et al. (2024); Lin et al. (2024) only tackle the initial distribution mismatch in the sampling process. Haber et al. (2025); Bojan et al. (2025) focus on the error correction of deterministic samplers. Our method focuses on a more general error correction scheme, including initial bias mismatch, discretization error, and score-estimation error with self-improving capability, and train- and test-time scalability, bringing significant performance gain.

## 3 PRELIMINARY

### 3.1 DIFFUSION MODELS

A diffusion model (Song et al., 2021b; Ho et al., 2020) consists of a *forward process* and a *reverse process*. The *forward process* gradually perturbs a *data distribution* $p^\star$ over $\mathbb{R}^d$ by injecting Gaussian noise. This process can be described by a *stochastic differential equation (SDE)*. For simplicity, we consider the *Ornstein-Uhlenbeck (OU) process*:

$$d\bar{\mathbf{x}}_t = -\bar{\mathbf{x}}_t\, dt + \sqrt{2}\, dB_t, \quad t \in [0, T], \quad \bar{\mathbf{x}}_0 \sim p^\star, \tag{1}$$

where $B_t$ is the standard Brownian motion. We use $\bar{p}_t$ to denote the probability density function (pdf) of the marginal distribution of $\bar{\mathbf{x}}_t$. The forward process in Eq. (1) converges exponentially fast to the standard Gaussian distribution $\mathcal{N}(0, I)$. It is associated with the following *reverse process*:

$$d\mathbf{x}_t = (\mathbf{x}_t + 2\nabla \log \bar{p}_{T-t}(\mathbf{x}_t))\, dt + \sqrt{2}\, dB_t, \quad t \in [0, T], \quad \mathbf{x}_0 \sim \bar{p}_T. \tag{2}$$

The term $\nabla \log \bar{p}_{T-t}(\cdot)$ is commonly referred to as the *score function* of the distribution $\bar{p}_{T-t}$. It is well known that the density $p_t$ of $\mathbf{x}_t$ generated by the reverse process in Eq. (2) coincides with that of the forward process, i.e. $p_t = \bar{p}_{T-t}$. Given an estimate $s_t(\cdot)$ of the score function $\nabla \log \bar{p}_{T-t}(\cdot)$, the reverse process can be approximated by:

$$d\mathbf{x}_t = (\mathbf{x}_t + 2s_t(\mathbf{x}_t))\, dt + \sqrt{2}\, dB_t, \ \mathbf{x}_0 \sim \mathcal{N}(0, I), \ t \in [0, T]. \tag{3}$$

We adopt the exponential integrator scheme (Zhang & Chen, 2023; Chen et al., 2023a) as a numerical SDE solver to simulate Eq. (3). We discretize the interval $[0, T]$ as $0, h, 2h, \ldots, T - h, T$ (ignore rounding issue). We initialize $\mathbf{x}_0 \sim \mathcal{N}(0, I)$ and calculate:

$$\mathbf{x}_{(k+1)h} = e^h \mathbf{x}_{kh} + 2s_{kh}(\mathbf{x}_{kh})(e^h - 1) + \sqrt{e^{2h} - 1}\, \eta_k, \quad \eta_k \sim \mathcal{N}(0, I), \tag{4}$$

for $k = 0, 1, \ldots, \frac{T}{h} - 1$. For convenience, we use $\hat{p}_t$ to denote the pdf of the distribution of $\mathbf{x}_t$ generated by the sampling process Eq. (4).

## 3.2 FLOW MATCHING

Flow matching (Lipman et al., 2023; Liu et al., 2022b) is another standard approach for sampling. It interpolates two distributions, $q_0$ and $q_1$, by an ordinary differential equation (ODE):

$$d\mathbf{z}_\tau = v(\mathbf{z}_\tau, \tau)\, d\tau, \quad \tau \in [0, 1] \tag{5}$$

where the velocity field $v(\cdot, \cdot)$ is the solution to the least squares problem:

$$\min_f \mathbb{E}_{Z_0 \sim q_0, Z_1 \sim q_1, \tau \sim \mathcal{U}([0,1])} \left[ \| (Z_1 - Z_0) - f(\tau Z_1 + (1-\tau)Z_0, \tau) \|_2^2 \right] \tag{6}$$

The interpolation means that: if Eq. (5) is initialized by a sample from the source distribution $\mathbf{z}_0 \sim q_0$, then the end point $\mathbf{z}_1$ follows the target distribution $q_1$. The source distribution is often chosen to be Gaussian, which is easy to sample from. In sampling tasks, one can learn a velocity field $\hat{v}(\cdot, \cdot)$ by minimizing equation 6 (with finite sample approximation) and simulate $d\mathbf{z}_\tau = \hat{v}(\mathbf{z}_\tau, \tau)\, d\tau$ by numerical ODE solvers.

## 4 METHODOLOGY

### 4.1 MOTIVATION

The diffusion model sampler Eq. (4) suffers from three main error sources Chen et al. (2023b;a):

- **Initial distribution mismatch:** the sampling process in Eq. (3) is initialized with standard Gaussian sample $\mathcal{N}(0, I)$, not $\bar{p}_T$;
- **Score estimation error:** the true score function $\nabla \log \bar{p}_{T-t}(\cdot)$ is approximated by $s_t(\cdot)$. Several factors (network approximation, finite sample, optimization) may contribute to the score estimation error.
- **Discretization error:** the process in Eq. (3) is further approximated by the numerical solver Eq. (4).

Standard predictor-corrector sampler (Song et al., 2021b) employs Langevin MCMC as a corrector to reduce the gap between the sampler and the ground truth:

$$d\mathbf{x}_\tau = s_t(\mathbf{x}_\tau) + \sqrt{2}\, dB_\tau, \quad \mathbf{x}_0 \sim \hat{p}_t. \tag{7}$$

However, the Langevin-based corrector has the following drawbacks:

- **Biased limiting distribution:** the (unnormalized) limiting distribution of Eq. (7) is $\exp(V)$ rather than $p_t$, where $\nabla V(\cdot) = s_t(\cdot)$ (Chewi et al., 2022). As a result, the error due to score approximation cannot be mitigated;

- **Slow mixing rate:** Langevin dynamics is known to have issues recovering the relative weights of different modes that are separated by low-density regions (Song & Ermon, 2019).

## 4.2 OUR APPROACH

We propose *Scalable Self-Improving Correction* (SSI-Corr), a self-improving method with train- and test-time scalability for diffusion models. Our approach introduces a flow-matching (Lipman et al., 2023; Liu et al., 2022b) subroutine within a sampling step as a corrector to mitigate the aforementioned errors. When applied at time step $t$, the corrector interpolates between $\hat{p}_t$, the biased marginal distribution generated by Eq. (4), and the ground-truth $p_t$.

The learning step for the corrector follows standard flow matching. We aim to minimize:

$$\min_{v \in \mathcal{V}} \mathbb{E}_{Z_0 \sim \hat{p}_t, Z_1 \sim p_t, \tau \sim \mathcal{U}([0,1])} \left[ \|(Z_1 - Z_0) - v(\tau Z_1 + (1 - \tau)Z_0, \tau)\|_2^2 \right]. \tag{8}$$

In practice, we simulate Eq. (4) up to time $t$ to sample from the distribution $\hat{p}_t$. To sample from the distribution $p_t$, we first sample from the data distribution $p^\star$ and insert Gaussian noise:

$$\mathbf{x} \sim p^\star, \quad Z_1 \sim \mathcal{N}(e^{-(T-t)}\mathbf{x}, (1 - e^{-2(T-t)})I).$$

Given sufficient samples from $\hat{p}_t$ and $p_t$, we can approximate the expectation in Eq. (8) by the empirical mean and learn a $\hat{v}$ via ERM.

To sample with the trained corrector $\hat{v}$, we simply run an additional ODE solver for $\hat{v}$ at time $t$ of the standard diffusion sampler Eq. (4) (see Algorithm 1).

Our SSI-Corr framework enjoys the following advantages:

- **Self-improving:** the improvement upon the base sampler depends only on $s_\cdot(\cdot)$, the base sampler itself, and the same data used to train the base sampler. No new data or external reward models are required.

- **Scalable dataset:** with sufficient computational resources, one can generate arbitrarily many samples from $\hat{p}_t$;

- **Training- and test-time scalability under different computational budgets:** as $t \to T$, the initial distribution mismatch, score estimation error, and discretization error accumulate (see Theorem 4.3). The joint error is reflected in the distance between $\hat{p}_t$ and $p_t$. Applying the corrector at timestep $t$ close to $T$ has the potential to mitigate all three errors. However, near $t = T$, both $\hat{p}_t$ and $p_t$ become multi-modal. As a result, the corrector velocity field $v$ is harder to learn, and learning and solving the ODE $d\mathbf{z}_\tau = v(\mathbf{z}, \tau) d\tau$ requires more computation. In contrast, the corrector has a simple near-linear form when $t = 0$ (see Theorem 4.4), requiring less computation for both training and testing. But when applied at $t = 0$, the corrector can only reduce the error due to the initial distribution mismatch, while errors from later steps remain.

- **Adaptation to other base sampler:** training the flow-matching corrector $\hat{v}$ is agnostic to the choice of base sampler, as long as samples from $\hat{p}_t$ and $p_t$ are available. In Section 5, we apply SSI-Corr to both DDPM (Ho et al., 2020) and DDIM (Song et al., 2021a).

---

**Algorithm 1** SSI-Corr Sampling

---

Inputs: pre-trained corrector $\hat{v}$, corrector time step $t$, diffusion sampler step size $h$, diffusion sampler time horizon $T$.
Sample initial state: $\mathbf{x}_0 \sim \mathcal{N}(0, I)$
**for** $k = 0, 1, \ldots, \frac{t}{h} - 1$ **do**
    **if** kh = t **then**
        Solve ODE: $\mathrm{d}\mathbf{z}_\tau = \hat{v}(\mathbf{z}_\tau, \tau)\,\mathrm{d}\tau, \ \tau \in [0, 1], \ \mathbf{z}_0 = \mathbf{x}_t$
        Set $\mathbf{x}_t = \mathbf{z}_1$
    **end if**
    $\eta_k \sim \mathcal{N}(0, I)$
    $\mathbf{x}_{(k+1)h} \leftarrow e^h \mathbf{x}_{kh} + 2s_{kh}(\mathbf{x}_{kh})(e^h - 1) + \sqrt{e^{2h} - 1}\eta_k$
**end for**
Output final sample $\mathbf{x}_T$

---

### 4.3 THEORETICAL ANALYSIS

**Error Analysis** In the sampling process, the corrector transfers samples from $\hat{p}_t$ to $\mathbf{z}_1$. If the flow matching corrector has small flow matching error in training step and small discretization error in sampling, the distribution of $\mathbf{z}_1$ can be closer to $p_t$ than $\hat{p}_t$, leading to an improvement upon the baseline sampler. We formalize this idea below.

We follow the standard assumptions in the diffusion model literature (Chen et al., 2023a;b):

**Assumption 4.1** (Score estimation error). *For all $0 \leq k \leq \frac{T}{h} - 1$,*

$$\mathbb{E}_{\mathbf{x}_{kh} \sim p_{kh}} \left[ \|s_{kh}(\mathbf{x}_{kh}) - \nabla \log p_{kh}(\mathbf{x}_{kh})\|_2^2 \right] \leq \epsilon_{\text{score}}^2.$$

**Assumption 4.2** (Smoothness). *For all $0 \leq t \leq T$, $\nabla \log p_t$ is $L$-Lipschitz.*

By adapting the theoretical result in Chen et al. (2023a), we get:

**Proposition 4.3.** *Let $\tilde{p}_t$ be the distribution of $\mathbf{z}_1$, and $\hat{p}$ be the distribution of the output of Algorithm 1. Suppose that Assumption 4.1 and 4.2 hold. If $L \geq 1$ and $h \leq 1$, then*

$$D_{\text{KL}}(p^\star || \hat{p}) \lesssim D_{\text{KL}}(p_t || \tilde{p}_t) + (T - t)\epsilon_{score}^2 + (T - t)dhL^2 \tag{9}$$

$D_{\text{KL}}(p_t || \tilde{p}_t)$ is the gap between the output of the corrector and the ground-truth marginal distribution. Because the target distribution of the corrector is $p_t$, the gap will be small when the corrector is trained and solved properly. The second term $(T - t)\epsilon_{\text{score}}^2$ and the third term $(T - t)dhL^2$ are the score matching error and discretization error induced by the base sampler Eq. (4) after the corrector step. Since we do not modify the base model, these errors cannot be reduced by the corrector.

For comparison, the sampling error without the corrector step is bounded by:

$$D_{\text{KL}}(p_t || \hat{p}_t) + (T - t)\epsilon_{\text{score}}^2 + (T - t)dhL^2,$$

where $D_{\text{KL}}(p_t || \hat{p}_t)$ is the gap between the base sampler and the ground truth at time $t$. It consists of the initial distribution mismatch, and the score matching error and the discretization error accumulated during time step $[0, t]$:

$$D_{\text{KL}}(p_t || \hat{p}_t) \lesssim O(e^{-T}) + t\epsilon_{\text{score}}^2 + tdhL^2,$$

When $D_{\text{KL}}(p_t || \tilde{p}_t) \leq D_{\text{KL}}(p_t || \hat{p}_t)$, the corrector pushes the base sampler closer to $p_t$ and effectively mitigate the error caused before that.

**Existence of a Simple Corrector** At the beginning of the sampling process, the gap between the sampler Eq. (4) and $p_t$ is dominated by $D_{\text{KL}}(p_0 || \mathcal{N}(0, I))$, the initial distribution mismatch. While applying the corrector at $t = 0$ does not help with the score matching error and discretization error, such a corrector has a nearly linear velocity field:

**Theorem 4.4.** *Let $v^\star$ be the ground-truth velocity field solving Eq. (8) when applied at $t = 0$. Let $M_2 := \mathbb{E}_{p^\star}[\|\mathbf{x}\|_2^2] \leq \infty$. If $T > 1$, then*

$$\int_0^1 \frac{\tau}{1 - \tau} \mathbb{E}_{\mathbf{z}_\tau} \left[ \left\| v^\star(\mathbf{z}_\tau, \tau) - \frac{2\tau - 1}{(1 - \tau)^2 + \tau^2} \mathbf{z}_\tau \right\|_2^2 \right] \mathrm{d}\tau = D_{\text{KL}}(p_0 || \mathcal{N}(0, I)) \lesssim (d + M_2)e^{-T},$$

*where $\mathbf{z}_\tau := (1 - \tau)\mathbf{z}_0 + \tau \mathbf{z}_1$. $\mathbf{z}_0 \sim \mathcal{N}(0, I)$ and $\mathbf{z}_1 \sim p_0 = \bar{p}_T$ are independent.*

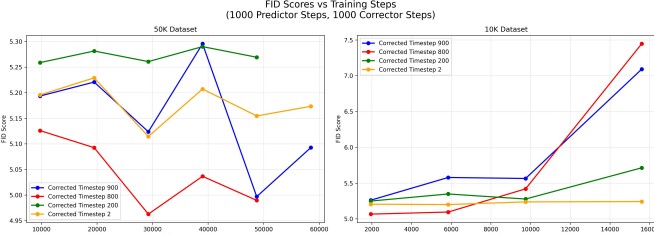 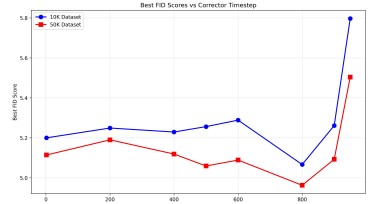

(a) This figure shows average FID scores (Y-Axis) across different training steps (X-Axis). The left figure shows the results of datasets of size 10K, and the right one for 50K. Different colors represent the timesteps at which the correctors are applied. The figure shows that the best FID score of timestep 2 saturates at around 5.2 with 10K training dataset and 9750 training steps. As for later timesteps, including 800 and 900, the FID scores continue to decrease until 4.9 while given a larger synthetic training dataset.

(b) This figure shows the best FID scores (Y-Axis) of 10K and 50K datasets across different corrector timesteps (X-Axis). Showing that later timesteps have a larger improving capacity due to more discretization error.

Figure 3: Training-time scalability comparison

Thus, the optimal corrector is approximately linear in expectation, meaning a simple model suffices to capture its behavior.

## 5 EXPERIMENTS

### 5.1 EXPERIMENT SETTINGS

Our experiments are conducted on Google Cloud Platform (GCP) with A2-Standard compute instance, accompanied with A100 GPUs (40G memory). For CIFAR10 dataset Krizhevsky (2009), we choose the original pre-trained DDPM on CIFAR10 with exponential moving average (EMA) training technique, released by Ho et al. (2020). For any dataset, we will first normalize the data to a standard normal distribution and train our correctors with AdamW optimizer Kingma & Ba (2015) (learning rate: 5e-4, weight decay: 0.0, $\beta_1$: 0.9, $\beta_2$: 0.999, $\epsilon$: 1e-8 ) and polynomial decay learning rate scheduler. The correctors are trained with 500 epochs and 256 batch size and same model architecture as the base model. We evaluate the model at (50, 100, 150, 200, 300, 400, 500) epochs and choose the best by FID scores. All FID scores reported are the average value across 3 runs with different random seeds and 10K .png samples.

In the following experiments, we will validate our claim in Section 1 correspondingly. First, in Section 5.2, we will present empirical evidence for the test-time scalability of our method. We will benchmark our method with baselines and evaluate the FID score with a growing compute budget at various correction timestep.

Secondly, Section 5.3 will validate the training-time scalability of our method. We visualize the best FID score versus different training budgets and correction timesteps. The experiment shows our method can scales well at the later timestep due to the accumulated discretization and score estimation errors while also providing the flexibility for the low compute budget at earlier correction timestep.

Finally, in Section 5.4, we investigate the generalizability of our method on the deterministic sampler, DDIM Song et al. (2021a), by visualizing the best FID scores across various correction timesteps. Our method can still offer additional FID improvement by 14.3 %, in comparison to the DDIM Song et al. (2021a) sampler with 1000 sampling steps.

### 5.2 TEST-TIME SCALABILITY

To validate the test-time scalability, we conduct two experiments in this section. Firstly, we compare our method with the standard predictor-corrector (PC) framework (Song et al., 2021b) and initial bias correction scheme as baselines. The result shows that our method surpasses the baseline method by 25.9% at least. Secondly, we provide the FID scores versus various correction timesteps with

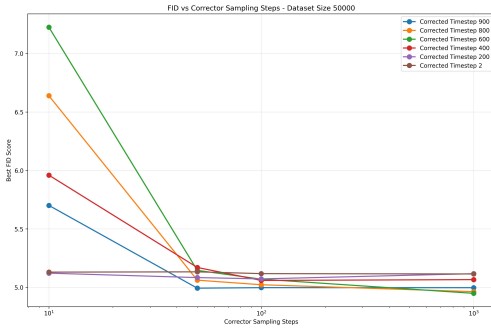 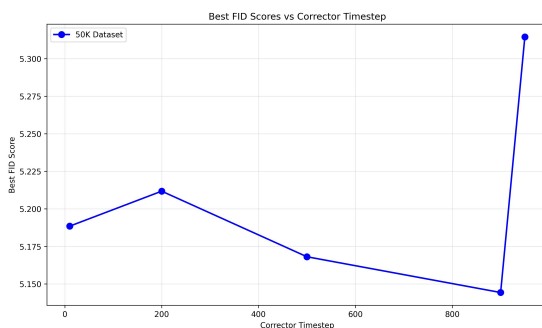

(a) This figure shows FID scores (X-Axis) versus the number of corrector sampling steps (Y-Axis). Each line represents a different corrected timestep. We can see the FID score with the corrector applied at timestep 2 and 200 are stable across different corrector sampling steps, while latter timesteps decrease significantly until 100 corrector sampling steps.

(b) The line plot demonstrates the best FID scores (Y-Axis) across different corrector timesteps (X-Axis) when using DDIM as the base sampler and 50K synthetic dataset. It illustrates that our method can get 14.3% than DDIM with 1000 sampling steps.

Figure 4: Fig. 3a shows the test-time scalability on different correction timesteps while Fig. 4b shows the training-time scalability with DDIM as the base sampler.

growing test-time compute budget, proving that the error can be mitigated with more compute budget at later correction timesteps.

**Compare with Baselines** The PC framework employs Langevin dynamics to tackle discretization error and initial distribution mismatch, while the initial bias corrector estimates the mean of $p_0$ and add bias to $\hat{p}_0 = \mathcal{N}(0, I)$. We follow the Song et al. (2021b) to implement PC methods with $N$ step Langevin dynamics as the corrector after every predictor (sampler) step. As for initial bias correction, we compute the average of the whole dataset as a naive bias correction to mitigate the initial distribution mismatch between training and inference. Then, we evaluate the PC sampler with various correction steps, including $N = 1, 2, 5, 10$. Both the base model and predictor of our method an PC sampler are DDPM and ancestral sampler with 1000 steps. In our framework SSI-Corr, the test tune scaling refers to the discretization steps to solve the ODE $d\mathbf{z}_\tau = \hat{v}(\mathbf{z}_\tau, \tau) \, d\tau$.

For each setting, we report the FID score with and without initial bias correction. The results are shown in the Figure 2 with total sampling steps (prediction plus correction steps) as X-axis and FID scores as Y-axis. Our method outperforms the PC sampler by more than 25% , regardless of the presence of bias correction and the number of correction steps. The PC sampler does not scale well with compute budgets: the FID scores do not improve much even with 10 times more sampling steps. It's even worse than DDIM Song et al. (2021a) with 100 sampling steps. Our observation coincides with the results in Song et al. (2021b). These results demonstrate the efficiency and scalability of our method over the baseline methods.

**Test-Time Scalability on Different Timesteps** To further illustrate the effect of the timestep choice on test-time scalability, we visualize the FID scores against corrector sampling steps when the correctors are applied at different correction step in Fig. 4a. The X-axis denotes the number of correction sampling steps. We fixed the base model and sampler as DDPM (pre-trained on CIFAR10) and ancestral sampler (Ho et al., 2020). As we can see in the figure, when applying the corrector at an earlier timestep, like $t = 2$, it has similar FID scores for different corrector sampling steps (5.161, with 10 corrector steps versus 5.114 with 1000 corrector steps). In contrast, the corrector applied at later correction timesteps, like $t = 900, 800$, benefit from increasing the corrector sampling steps (suboptimal FID scores 6.535 and 7.111 with 10 corrector sampling steps versus 5.093 and 4.963 1000 corrector sampling steps. This result shows excellent test-time scalability of our corrector when applied at later sampling steps (e.g. $t = 800, 900$) and computational efficiency when applied at earlier steps. We present further numerical results in the appendix.

## 5.3 TRAINING-TIME SCALABILITY

To demonstrate the training scalability of our method, we conduct two experiments.

The first one shows the best FID scores with various timesteps and trajectory dataset sizes, i.e., number of samples from $\hat{p}_t$. It showcases a larger improvement when the corrector is applied at a later correction timestep at the cost of more computation.

The second experiment shows that the FID scores evolve with training steps with different correction timesteps. It demonstrates that our corrector at earlier timesteps is a computationally efficient way to reduce FID scores significantly.

These two experiments demonstrate the flexibility of our method under various computational budgets. Both experiments support the claim that our method offers training scalability at later timesteps: the FID scores go down as the computational budget increases. On the other hand, earlier correction timesteps yield a better computational efficiency at training time.

**Training-Time Scalability across Correction Timesteps and Training Trajectory Sizes** The Fig. 3b shows the improvement of the best FID score for each correction timestep from 10K to 50K training trajectories (i.e. training samples from $\hat{p}_t$) as the empirical evidence for training-time scalability. The X-axis is the correction timestep and the Y-axis is the FID scores. As the figure Fig. 3b shows, the FID scores of 50K go down faster than 10K as the correction timestep goes up. The improvement gap from 10K to 50K grows as the correction timestep increases, except for the correction timestep 800. Therefore, it shows strong evidence that the later correction timesteps yield more improved capacity as the discretization and score matching errors accumulate.

We also notice that the FID scores go up quickly after the correction timestep 800. It is because the target distribution becomes even less smooth, and the corrector is harder to train.

**Training-Time Scalability across Correction Timesteps and Training Steps** In addition, we also plot the FID scores at different training step in Fig. 3a. The X-axis is the training step while Y-axis is the FID scores. We synthesize 10K (right) and 50K (left) trajectories and train correctors on them. As we can see, no matter how large the dataset is or how many training step the model has taken, the FID scores of timestep 2 always remain stable between $5.2$ to $5.15$. It shows that training a corrector at early timesteps can reduce the FID scores with very limited computational resource. In contrast, the later timestep, like 800 and 900, will overfit on the small training set, which represents the higher demand of computational budget (for training data generation) to mitigate more discretization and score estimation errors. In conclusion, the experiment shows that the optimal correction timestep choice varies with the given compute budget. The later timesteps offer larger improving capacity while requiring more computational resources. On the other hand, earlier timesteps provide limited performance gain but give a better training-performance-gain ratio.

### 5.4 GENERALIZABILITY TO VERSATILE BASE SAMPLERS

Here, we further demonstrate the generalizability of our method to deterministic samplers, like DDIM Song et al. (2021a). We choose DDIM with 1000 sampling step as the base sampler and follow the same training and evaluation protocol as in Section 5.2. As the results shown in Fig. 4b, our method achieve FID score $5.14$ at timestep 900 with 50K synthetic dataset, surpassing the FID score of DDIM sampler $6.0$ with 1000 sampling steps. The results show that our method can adapt to deterministic or further more samplers and provides significant performance improvement by 14.3%.

## 6 CONCLUSION

In the paper, we have demonstrated the scalability and self-improving capability of our method for improving DMs without the generation-verification framework. Our methods are motivated by the error analysis insights. Our empirical results show a 27% improvement on the FID score, demonstrating the strong self-improving capacity and scalability of our method. We also call for more investigation into self-improving methods with theoretical guarantees.

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
