APPENDIX

## A  PROOF DETAILS

**Lemma A.1.** *Let* $\mathbf{z}_0 \sim \mathcal{N}(0, I)$, $\mathbf{z}_1 \sim Q$ *be two independent random vectors. Let* $t \in [0, 1]$ *and* $\mathbf{z}_t := (1 - t)\mathbf{z}_0 + t\mathbf{z}_1$. *Then*

$$\mathbb{E}[\mathbf{z}_1 - \mathbf{z}_0|\mathbf{z}_t] = \frac{1 - t}{t}\nabla_{\mathbf{z}_t} \log q(\mathbf{z}_t) + \frac{1}{t}\mathbf{z}_t,$$

*where* $q(\mathbf{z}_t)$ *is the pdf of the marginal distribution of* $\mathbf{z}_t$.

*Proof.* Because

$$\mathbf{z}_0 = \frac{1}{1 - t}\mathbf{z}_t - \frac{t}{1 - t}\mathbf{z}_1,$$

we have

$$\mathbb{E}[\mathbf{z}_1 - \mathbf{z}_0|\mathbf{z}_t] = \mathbb{E}[\mathbf{z}_1 - \frac{1}{1 - t}\mathbf{z}_t + \frac{t}{1 - t}\mathbf{z}_1|\mathbf{z}_t] = \frac{1}{1 - t}\mathbb{E}[\mathbf{z}_1|\mathbf{z}_t] - \frac{1}{1 - t}\mathbf{z}_t.$$

By definition,

$$\mathbf{z}_t|\mathbf{z}_1 \sim \mathcal{N}(t\mathbf{z}_1, (1 - t)^2 I).$$

Then the score function of $\mathbf{z}_t|\mathbf{z}_1$ is:

$$\nabla_{\mathbf{z}_t} \log q(\mathbf{z}_t|\mathbf{z}_1) = -\frac{\mathbf{z}_t - t\mathbf{z}_1}{(1 - t)^2}.$$

Thus

$$\mathbf{z}_1 = \frac{(1 - t)^2}{t}\nabla_{\mathbf{z}_t} \log q(\mathbf{z}_t|\mathbf{z}_1) + \frac{1}{t}\mathbf{z}_t$$

Thus we can further rewrite the conditional expectation as:

$$\mathbb{E}[\mathbf{z}_1 - \mathbf{z}_0|\mathbf{z}_t] = \frac{1}{1 - t}\mathbb{E}[\mathbf{z}_1|\mathbf{z}_t] - \frac{1}{1 - t}\mathbf{z}_t$$

$$= \frac{1 - t}{t}\mathbb{E}[\nabla_{\mathbf{z}_t} \log q(\mathbf{z}_t|\mathbf{z}_1)|\mathbf{z}_t] + \frac{1}{t(1 - t)}\mathbf{z}_t - \frac{1}{1 - t}\mathbf{z}_t$$

$$= \frac{1 - t}{t}\int \nabla_{\mathbf{z}_t} \log q(\mathbf{z}_t|\mathbf{z}_1)q(\mathbf{z}_1|\mathbf{z}_t)\,\mathrm{d}\mathbf{z}_1 + \frac{1}{t}\mathbf{z}_t$$

$$= \frac{1 - t}{t}\int \frac{\nabla_{\mathbf{z}_t} q(\mathbf{z}_t|\mathbf{z}_1)}{q(\mathbf{z}_t|\mathbf{z}_1)}\frac{q(\mathbf{z}_t|\mathbf{z}_1)q(\mathbf{z}_1)}{q(\mathbf{z}_t)}\,\mathrm{d}\mathbf{z}_1 + \frac{1}{t}\mathbf{z}_t$$

$$= \frac{1 - t}{t}\frac{\nabla_{\mathbf{z}_t}\int q(\mathbf{z}_t|\mathbf{z}_1)q(\mathbf{z}_1)\,\mathrm{d}\mathbf{z}_1}{q(\mathbf{z}_t)} + \frac{1}{t}\mathbf{z}_t$$

$$= \frac{1 - t}{t}\frac{\nabla_{\mathbf{z}_t} q(\mathbf{z}_t)}{q(\mathbf{z}_t)} + \frac{1}{t}\mathbf{z}_t$$

$$= \frac{1 - t}{t}\nabla_{\mathbf{z}_t} \log q(\mathbf{z}_t) + \frac{1}{t}\mathbf{z}_t.$$

$\square$

## A.1  PROOF OF THEOREM 4.4

*Proof.* We consider the following two flow matching models:

$$\mathrm{d}\mathbf{z}_\tau = v^\star(\mathbf{z}_\tau, \tau)\,\mathrm{d}\tau, \quad \tau \in [0, 1]$$

$$\mathrm{d}\mathbf{z}_\tau = v^G(\mathbf{z}_\tau, \tau)\,\mathrm{d}\tau, \quad \tau \in [0, 1],$$

where $v^\star$ interpolates $\mathcal{N}(0, I)$ and $p_0$, $v^G$ interpolates $\mathcal{N}(0, I)$ and $\mathcal{N}(0, I)$, both with independent coupling. We use $q_\tau^\star$ to denote the marginal distribution of $\mathbf{z}_\tau$ generated by $v^\star$ and use $q_\tau^G$ to denote the marginal distribution of $\mathbf{z}_\tau$ generated by $v^G$.

Specifically, we have $v^\star(\mathbf{z}_\tau, \tau) := \mathbb{E}[\mathbf{z}_1 - \mathbf{z}_0 | \mathbf{z}_\tau]$ for $\mathbf{z}_\tau := (1-\tau)\mathbf{z}_0 + \tau\mathbf{z}_1$, $\mathbf{z}_0 \sim \mathcal{N}(0, I) \perp \mathbf{z}_1 \sim p_0$; $v^{\mathrm{G}}(\mathbf{z}_\tau, \tau) := \mathbb{E}[\mathbf{z}_1 - \mathbf{z}_0 | \mathbf{z}_\tau]$ for $\mathbf{z}_\tau := (1 - \tau)\mathbf{z}_0 + \tau\mathbf{z}_1$, $\mathbf{z}_0 \sim \mathcal{N}(0, I) \perp \mathbf{z}_1 \sim \mathcal{N}(0, I)$.

By Lemma A.1, we have:

$$v^\star(\mathbf{z}_\tau, \tau) = \frac{1 - \tau}{\tau} \nabla_{\mathbf{z}_\tau} \log q_\tau^\star(\mathbf{z}_\tau) + \frac{1}{\tau}\mathbf{z}_\tau$$

$$v^{\mathrm{G}}(\mathbf{z}_\tau, \tau) = \frac{1 - \tau}{\tau} \nabla_{\mathbf{z}_\tau} \log q_\tau^{\mathrm{G}}(\mathbf{z}_\tau) + \frac{1}{\tau}\mathbf{z}_\tau$$

By arranging, we have:

$$\nabla \log \frac{q_\tau^\star(\mathbf{z}_\tau)}{q_\tau^{\mathrm{G}}(\mathbf{z}_\tau)} = \frac{t}{1 - t} \left( v^\star(\mathbf{z}_\tau, \tau) - v^{\mathrm{G}}(\mathbf{z}_\tau, \tau) \right).$$

Applying Lemma 6 of Chen et al. (2023), we get:

$$\frac{\partial}{\partial \tau} D_{\mathrm{KL}}(q_\tau^\star || q_\tau^{\mathrm{G}}) = \mathbb{E}_{\mathbf{z}_\tau \sim q_\tau^\star} \left[ \left\langle v^\star(\mathbf{z}_\tau, \tau) - v^{\mathrm{G}}(\mathbf{z}_\tau, \tau), \nabla \log \frac{q_\tau^\star(\mathbf{z}_\tau)}{q_\tau^{\mathrm{G}}(\mathbf{z}_\tau)} \right\rangle \right]$$

$$= \frac{\tau}{1 - \tau} \mathbb{E}_{\mathbf{z}_\tau \sim q_\tau^\star} \left[ \left\| v^\star(\mathbf{z}_\tau, \tau) - v^{\mathrm{G}}(\mathbf{z}_\tau, \tau) \right\|_2^2 \right].$$

Integrate over $\tau \in [0, 1]$ on both sides:

$$D_{\mathrm{KL}}(q_1^\star || q_1^{\mathrm{G}}) - D_{\mathrm{KL}}(q_0^\star || q_0^{\mathrm{G}}) = \int_0^1 \frac{\tau}{1 - \tau} \mathbb{E}_{\mathbf{z}_\tau \sim q_\tau^\star} \left[ \left\| v^\star(\mathbf{z}_\tau, \tau) - v^{\mathrm{G}}(\mathbf{z}_\tau, \tau) \right\|_2^2 \right] \mathrm{d}\tau. \tag{1}$$

Because $q_0^\star = q_0^{\mathrm{G}} = \mathcal{N}(0, I)$, $D_{\mathrm{KL}}(q_0^\star || q_0^{\mathrm{G}}) = 0$. By Lemma 9 of Chen et al. (2023), $D_{\mathrm{KL}}(q_1^\star || q_1^{\mathrm{G}}) = D_{\mathrm{KL}}(\bar{p}_T || \mathcal{N}(0, I)) \lesssim (d + M_2) e^{-T}$.

By definition, we have $q_\tau^{\mathrm{G}} = \mathcal{N}(0, (\tau^2 + (1 - \tau)^2)I)$, thus

$$v^{\mathrm{G}}(\mathbf{z}_\tau, \tau) = -\frac{1 - \tau}{\tau} \frac{\mathbf{z}_\tau}{\tau^2 + (1 - \tau)^2} + \frac{1}{\tau}\mathbf{z}_\tau = \frac{2\tau - 1}{(1 - \tau)^2 + \tau^2}\mathbf{z}_\tau.$$

We finish the proof by plugging these into Eq. (1).

$\square$

# B  DETAILED FID SCORES

We additionally provide FID scores for each correction timestep with best training epoch in Table 1 and Table 2. We generate samples by 1000 prediction and 1000 correction steps. For each timestep, we sample 50K trajectories dataset, train the correctors for 500 epochs, and choose the best epoch from (50, 150, 100, 150, 200, 300, 400, 500).

| Corrected Timestep | Best Epoch | Corrector Sampling Steps | | |
|---|---|---|---|---|
| | | 10 | 100 | 1000 |
| 2 | 150 | 5.217 | 5.211 | 5.200 |
| 200 | 50 | 5.209 | 5.245 | 5.249 |
| 400 | 150 | 6.142 | 5.249 | 5.229 |
| 500 | 50 | 7.340 | 5.250 | 5.256 |
| 600 | 50 | 8.151 | 5.269 | 5.289 |
| 800 | 50 | 6.614 | 5.119 | 5.067 |
| 900 | 50 | 5.868 | 5.239 | 5.261 |
| 950 | 150 | 5.756 | 5.648 | 5.797 |

Table 1: FID Scores vs Corrector Sampling Steps - Dataset Size 10000

| Corrected Timestep | Best Epoch | Corrector Sampling Steps | | | |
|---|---|---|---|---|---|
| | | 10 | 50 | 100 | 1000 |
| 2 | 150 | 5.161 | 5.208 | - | 5.114 |
| 200 | 400 | 5.172 | 5.164 | 5.243 | 5.190 |
| 400 | 400 | 6.449 | 5.170 | 5.251 | 5.119 |
| 500 | 400 | 8.245 | 5.327 | 5.221 | 5.059 |
| 600 | 300 | 8.865 | 5.399 | 5.216 | 5.089 |
| 800 | 150 | 7.111 | 5.164 | 5.073 | 4.963 |
| 900 | 300 | 6.535 | 5.143 | 5.145 | 5.093 |
| 950 | 200 | 5.604 | 5.310 | 5.376 | 5.504 |

Table 2: FID Scores vs Corrector Sampling Steps - Dataset Size 50000

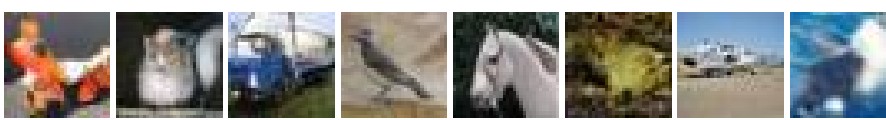

Figure 1: Timestep 900.

## C  VISUALIZATION

Here, we visualize CIFAR10 samples generated by our corrector method at various timesteps. The training and sampling details are the same as Section B. We visualize samples at correction timestep 900, 800, 600, 500, 400, 200, 100, and 2 in Fig. 1, Fig. 2, Fig. 3, Fig. 4, Fig. 5, Fig. 6, Fig. 7, and Fig. 8 respectively.

## REFERENCES

Hongrui Chen, Holden Lee, and Jianfeng Lu. Improved analysis of score-based generative modeling: User-friendly bounds under minimal smoothness assumptions. In *International Conference on Machine Learning*, pp. 4735–4763. PMLR, 2023.

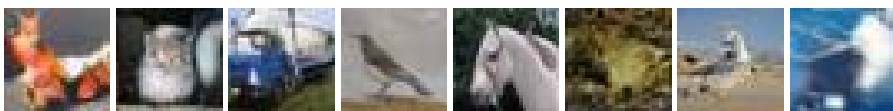

Figure 2: Timestep 800.

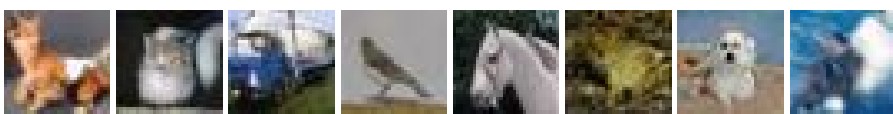

Figure 3: Timestep 600.

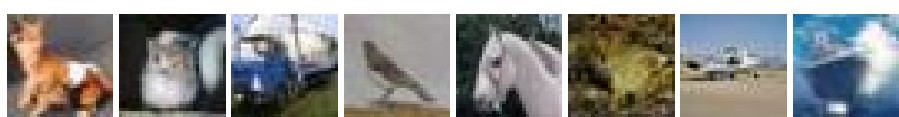

Figure 4: Timestep 500.

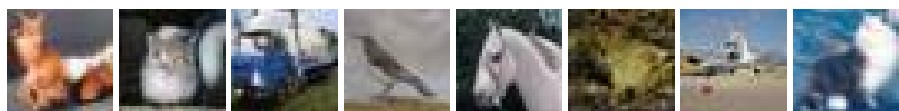

Figure 5: Timestep 400.

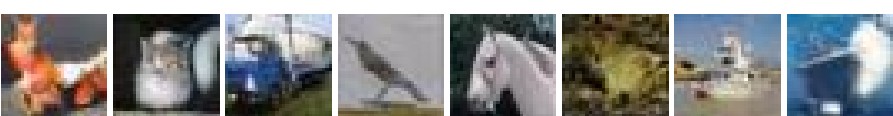

Figure 6: Timestep 200.

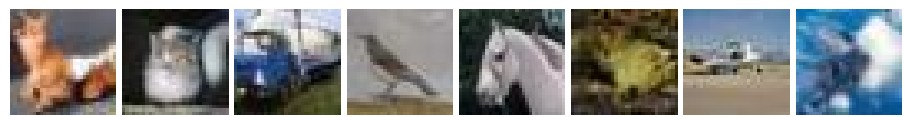

Figure 7: Timestep 100.

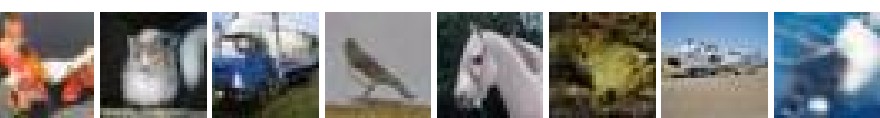

Figure 8: Timestep 2.