# OpenReview forum: "Test Time Scaling of Diffusion Model via Flow Matching Corrector"
_ICLR.cc/2026/Conference — Submitted to ICLR 2026_

### Official Review · Reviewer_2zaJ · 2025-10-28

**Soundness:** 2
**Presentation:** 2
**Contribution:** 2
**Rating:** 4
**Confidence:** 4

**Summary:**

The paper proposes SSI-Corr, a “self-improving” framework for diffusion models that inserts a flow-matching corrector at a chosen timestep  to transport samples from the sampler’s biased marginal  toward the target marginal . Practically, the corrector is trained with synthetic trajectories from the base sampler and noisy data pairs constructed from the original dataset; at inference, an ODE for the learned velocity field is solved at step . The authors claim test-time and training-time scalability and report CIFAR-10 FID improvements for DDPM (e.g., from 6.872 to 4.963 by roughly doubling total steps) and a 14.3% gain over DDIM. They provide an error decomposition and a KL bound arguing that reducing  at step  should improve the final sample distribution.

**Strengths:**

Clear formulation that decouples correction from the base sampler and can be slotted at arbitrary timesteps; the method is conceptually simple (one extra ODE solve).

Theory-backed motivation: a clean error analysis (initial bias, score error, discretization) and a KL bound illustrating when the corrector helps; also a note that the  corrector is near-linear in expectation.

**Weaknesses:**

1. Limited scope of evaluation: only unconditional CIFAR-10 is shown. No higher-resolution datasets (e.g., ImageNet-64/256, CelebA-HQ) or text-to-image settings, making scalability/generalization claims hard to assess.

2. Self-improvement framing is overstated: the method still uses the original dataset to construct targets ; it is not data-free (just no new data). The paper should temper the claim and discuss limitations when access to data is restricted.

3. Baselines feel weak/incomplete: comparison is mainly to predictor–corrector (Langevin) and an initial-bias tweak; no head-to-head against modern fast samplers / correctors (e.g., DPM-Solver variants, UniPC, consistency models, or recent flow-matching refinements) under matched compute. The fairness of comparing “one-shot” correction at a single  to PC applied after every step is also not fully justified.

4. Compute accounting is unclear: the principal gains come when total sampling steps roughly double; wall-clock impact of the extra ODE solve, solver tolerances, and memory footprint are not reported. Training cost vs. improvement (especially at large ) also lacks a thorough cost–benefit analysis.

**Questions:**

see weakness

---

> ### Author Response · Authors · 2025-11-26
> **Response to Review 2zaJ**
>
> # Large Dataset
>
> Thank you for your suggestion. We are currently working on CelebA-HQ. We will update our results as soon as possible.
>
> # Self-Improving Overstate
>
> Thank you for the valuable suggestion. However, we need to clarify that the self-improving procedure doesn't exclude the training dataset from both theoretical and pratical perspectives. To elaborate the pratical scenario, refer to section 3.3 of the paper, LARGE LANGUAGE MODELS CAN SELF-IMPROVE, which is the one of the most important paper proposing self-improving language model, the paper use a few existing human written questions (however, they don't clarify if the questions are inside the training dataset of the language model or not) to prompt the language models to generate more diversed questions for self-improving training. Thus, even for the self-improving language model, they still use some ground-truth data to do data synthesis and self-improving.
>
> From theoretical perspective, the paper, Mind the Gap: Examining the Self-Improvement Capabilities of Large Language Models, defines the response generation as: "Let $\mathcal{X}$ be the prompt / context space, and $\mathcal{Y}$ be the response space. Let $\mathcal{F} \subseteq \{f: \mathcal{X} \to \Delta (\mathcal{Y})\}$ be a class of generative models that maps a prompt to a distribution over responses". The paper doens't exlude the training or ground-truth dataset from data synthesis process. As a result, from both theoretical and pratical perspectives, these two papers shows that the self-improving procedure does laverage in-distribution samples to synthesize a new training dataset for self-improving, which aligns with our setting.
>
> # Self-Improving with Small Training and Synthetic Dataset
>
> Furthermore, we've also conducted an experiment to show the training scalability of our method with small training and synthetic dataset. Here, we show the FID score trained with only 10K trtaining and synthetic dataset, as we can see, we can still achieve FID 5.15 at earlier correction timestep. We claim our method is a strong self-improving method that bring significant FID improvement even with small training dataset.
>
> | **Corr Timestep** | **FID**              |
> |-------------------|----------------------|
> | 200               | 5.217 ± 0.110        |
> | 2                 | 5.150 ± 0.043        |
>
> # Baselines Comparison
>
> Thank you for your comment. Here is the FID scores of UniPC, DPM-Solver, and UniPC with 1000 sampling steps and 10K samples for computing FID score. As we can see the state-of-the-art training-free samplers cna only achieve at most 6.02 FID score, which is much worse than our method.
>
> | **Corr Timestep**      | **FID**              |
> |------------------------|----------------------|
> | PNDM                   | 6.29                 |
> | DPM-Solver++ (Order 2) | 6.08                 |
> | UniPC                  | 6.02                 |
>
>
> # Computing Budget Accounting
>
> Thank you for your valuable advice. Here we provide the training cost details. We've remove the evaluation and pre-trained CIFAR10 DDPM from the training script and only account the foward and backword pass of training corrector. We frozen 95% of layers for fine-tuning and train correctoir with learning rate 1e-4 for 100 epochs, while use 5e-4 as learning rate and 100 epochs for non-fine-tuning method. As we can see, with fine-tuning method, we can reduce about 30% memory footprint, 95% trainable parameters, and 39% training time.
>
> | **W / WO Fine Tuning** | **GPU Memory** | **Training Time** |
> |---------------------|----------------------|--------|
> | With Fine-Tuning    | 11.17 GB (11,444MiB) | 3556s  |
> | Without Fine-Tuning | 15.99 GB (16,374MiB) | 5870s  |
>
> In comparison to pre-training a diffusion model and flow matching model, which takes about 800 and 500 epoch respectively, our method is a light-weight module costs only 5% training parameter and 12.5% to 20% training iteration.

---

### Official Review · Reviewer_4TfU · 2025-10-30

**Soundness:** 3
**Presentation:** 3
**Contribution:** 2
**Rating:** 2
**Confidence:** 4

**Summary:**

This paper introduces the Scalable Self-Improving Correction (SSI-Corr) method to address three intrinsic sources of error in diffusion models: initial distribution mismatch, score estimation error, and discretization error.
The proposed approach trains an additional model to bridge the gap between the true distribution (ground truth) and the model distribution (obtained from the base diffusion sampler) at a specific diffusion timestep $t$. This corrector model is trained using the well-known flow matching loss.
The paper theoretically establishes an explicit bound on the sampling error of this method and empirically demonstrates its effectiveness through experiments.

**Strengths:**

- The paper proposes a principled approach based on flow matching to mitigate key sources of error inherent to diffusion models.

- It is not restricted to a specific sampler and can be applied to any diffusion sampling process, regardless of the underlying architecture or parameterization.

- It provides strong theoretical foundations, clearly deriving the specific error bounds achieved by the proposed algorithm.

**Weaknesses:**

- The method introduces an additional ODE solving step, inevitably increasing the total sampling time.

Moreover, since the ODE solver uses the same architecture as the base score model, it is computationally heavy.

Even for a relatively simple setup such as CIFAR-10 with a lightweight score model, the proposed framework requires an additional model of equal size.

Thus, it is unclear whether the method can be scalably applied to large and powerful diffusion models [1,2]. [2] has cifar-10 experiment.

- Experiments are conducted only on small-scale datasets, and comparisons are made mainly with older baselines.

It remains uncertain whether the proposed method would still be effective when applied to larger datasets or more recent diffusion architectures beyond DDPM and CIFAR-10.

-----------
[1] Scaling Rectified Flow Transformers for High-Resolution Image Synthesis

[2] Direct Discriminative Optimization: Your Likelihood-Based Visual Generative Model is Secretly a GAN Discriminator

**Questions:**

- In Figure 3(a), the FID score appears to increase as training progresses, suggesting a lack of convergence.

This raises concerns about whether the proposed method is actually improving the sample quality as intended.

The observed instability might be due to the non-optimal training of the corrector at specific timesteps, but a more detailed explanation would be helpful.

- The proposed method appears general enough to be applicable to distillation models as well.

Do the authors have any plans for additional experiments or evaluations in such settings?

---

> ### Author Response · Authors · 2025-11-26
> **Response to Review 4TfU**
>
> # Heavy Training Cost and Training Stability
>
> Thank you so much for your insightful advice. Our corrector is harder to train than regular flow matching or diffusion models because it requires more sophisticate flow learning. Therefore, we would like propose a simple re-initialization and fine-tuning trick to stablize the training and reduce 95\% training cost. We freeze the pre-trained DDPM by 95\% and fine-tune it with 1e-4 learning rate on 10K synthetic dataset. For each correction timestep, we repeated the experiment by 3 times and report the average FID across 3 runs. We report the final FID scores for correction timesteps on 800, 200, and 2.
>
> | **Corr Timestep** | **FID**              |
> |-------------------|----------------------|
> | 800               | 5.021 ± 0.051        |
> | 200               | 5.133 ± 0.020        |
> | 2                 | 5.186 ± 0.088        |
>
> Also, here is the FID score, training loss, and validation loss during training,  (https://imgur.com/a/lQe8rlw). As we can see, both FID and losses decrease stably during training. Note that the timestep shown in the figure is the complementary of Corr Timestep of **Corr Timestep** shown in the table, which means "corr_trained_timestep: 200" in the figure corresponds to "Corr Timestep 800" in the table, "corr_trained_timestep: 800" represents "Corr Timestep: 200", and "corr_trained_timestep: 998" in figure should be interpreted "Corr Timestep: 2" in table.
>
> With such simple re-initialization and fine-tuning trick, we can achieve similar performance as original paper with only 5\% trainable parameters stably. It prove that our method is a reliable light-weight module to achieve self-improving for diffusion model.
>
> # Large Scale Dataset
>
> Thank you for your awesome advice. We are working on CelebA-HQ currently. We will update our results as soon as possible.
>
> # Applying on More Advanced Diffusion Models
>
> Thank you for your valuable review. We will apply our method on consistency models. We will report our results as soon as possible.
>
> # We Target Different Scenarios from Diffusion Model Distillation
>
> Here we want to clarify the motivation of our work again. The existing diffusion distillaiton method, including consistency model and rectflied flow focus on improving the FID score under low sampling steps. Their method does really improve the image quality generated by a few step to become competitive with ones generated by 1000 steps. However, such distillation suffers from 2 issues, (1) The image quality cannot surpass original model before distillation (2) The image quality doesn't grow along with the increasing sampling steps, sometime it even get worse.
>
> To tackle these 2 issues, we propose a self-improving correction method that can bring improvement of image quality and also keep inference-time scaling. Our experiment also prove our claim.

---

### Official Review · Reviewer_dJp8 · 2025-10-31

**Soundness:** 1
**Presentation:** 1
**Contribution:** 1
**Rating:** 2
**Confidence:** 4

**Summary:**

The authors propose to improve the generative modelling of diffusion models post-training by learning an auxiliary flow matching model that maps the generated distribution to the true data distribution. They train this auxiliary model in practice using the original training data and synthetic data generated by the diffusion model *at a specific noise level*.  At inference, the flow matching model is used at the aforementioned noise level to "correct" the generated samples, diffusion generation then continues as normal.

**Strengths:**

I believe the core idea is reasonable and may have potential in practical generative modelling applications with diffusion models. This sort of approach may be interesting to explore in test-time distribution shifting (e.g. adjusting the generative style of a frozen/black box model post-training).

**Weaknesses:**

The paper load for ICLR this year has been large, and so I have not been able to spend as much time as I would like on reviewing. I encourage the authors to correct any errors/misunderstandings I may have with regards to the paper.

1. **Understanding the approach**
    1. The approach applies a very general interpolation between the generated and data distribution via random pairing at train time, which to me would potentially suggest large transformations during the correction step. However, the actual effect (Figure 1) seems to be visually imperceptible. The authors do not provide any insight on why this is the case. Why does the learnt velocity make only very small changes?
2. **Poor presentation**
    1. Figures are out of order, e.g. Figure 4 appears in the text before Figure 3, leading to confusion when reading.
    1. Figures are poorly formatted with text inside figures being tiny and close to illegible.
    1. Writing is unpolished e.g. in the introduction, "current works ... either rely on additional reward models for guidance."
3. **Weak experiments**
    1. The authors only benchmark on CIFAR-10. That is to say a single, small-scale (32x32) dataset. I am not saying that CIFAR is necessarily a bad benchmark for generative modelling, however, using *only* CIFAR-10 is not enough. This is especially problematic since the authors motivate their work from the perspective of test-time scaling, which is a concept that is primarily relevant in large-scale foundation model deployment, where relevant experiments should be at the scale of high-resolution text-to-image generation.
    1. The lack of any FID-50K results makes it hard to compare the results to the literature.

**Questions:**

See weaknesses

---

> ### Author Response · Authors · 2025-11-26
> **Resonse to Review dJp8**
>
> # Our Novelty and Contribution
>
> Thank you for your valuable comment. Our method targets on proposing a brand new self-improving framework for diffusion models based on ODE error analysis. Based on our analysis, we develop a light-weight fined-tuned corrector can provide significant error correction for the velocity field of the diffusion models. Althought the figure 1 is visually imperceptible by naked eyes because the FID score of diffusion model is very low and our corrector deomonstrate strong capability on correcting non-trivial error on velocity field, our method still yield great improvement on FID scores (10K samples) from 6.7 (DDIM 1000 steps) to 4.9 (our method). We need to emphasize that the visually imperceptible issue of figure is not because of the limitation of our method, in contrast, it demonstrates that our method is very powerful to correct small and imperceptible error, tackle the bottleneck of diffusion model itself.
>
> # Larger Dataset
>
> Thank you for your insightful suggestion. Due to the computational limitation, we're working hard on the CelebA-HQ. We will update our FID results as soon as possible. However, we believe our CIFAR10 result is a convincing protoype for our method.
>
> # Compute FID with 50K Samples
>
> Thank you your great suggestion. Due to the computational limitation, sampling 50K from diffusion model would be hard for us. Here, we provide our FID score with 50K samples. We sample 50K synthetic dataset and freeze 95% of the model with 1e-4 learning rate and 100 epochs. The FID result is shown as following. Applying corrector at tiemstep 800 still offers the best FID score, 2.9 . Note that This strong result demonstrate our method is very competitve to other strong diffusion methods.
>
> | **Corr Timestep** | **FID**              |
> |-------------------|----------------------|
> | 800               | 2.928                |
> | 200               | 3.137                |
>
> However, we believe the results of 10K sample is sufficient to demonstrate that our method yields strong improvement in comparison to the baseline predictor-corrector methods and DDIM, which reduce the FID score from 6.7 to 4.9, about 27% improvement.
>
> # Presentaiton Improvement
>
> Thank you for your carful review. We will correct the figure caption in figure 4 and further polish our writing. We will also add one more paragraph to emphasize how our method can bring self-improving and how our method different from existing distillation methods.

---

### Official Review · Reviewer_TZ1Z · 2025-10-31

**Soundness:** 2
**Presentation:** 2
**Contribution:** 2
**Rating:** 4
**Confidence:** 3

**Summary:**

The paper introduces Scalable Self-Improving Correction (SSI-Corr), a drop-in framework for diffusion models that replaces generation–verification schemes with a flow-matching corrector trained to align the sampler’s marginal with the ground-truth marginal at a chosen timestep t. The authors provide a clean error decomposition and a bound showing that the final divergence is controlled by the corrector’s mismatch at t plus residual score/discretization terms accumulated after t. Empirically, SSI-Corr improves FID on a pre-trained CIFAR-10 DDPM by ~27% with the ancestral sampler and ~14.3% with DDIM, and exhibits both test-time and training-time scalability via the choice of correction timestep and solver steps.

**Strengths:**

- The proposed method is simple and easy to plug in to the current diffusion models.
- The introduction of flow matching to the self-improving corrector is new
- Several experiments are conducted to demonstrate the main claims
- Implementation and evaluation details are reported clearly enough to replicate the results.

**Weaknesses:**

- Incremental core novelty. The main technical part—using flow matching to train a corrector at a chosen timestep—leans on established techniques; much of the machinery (least-squares flows, OU marginals, sampler integration) is also adapted from prior work.
- The writing could be improved; for example, the discussion of training and testing scalability appears before these terms are clearly defined.
- The proposed corrector can overfit training data (the paper notes this qualitatively). Please quantify with train/val curves vs dataset size, and report generalization across seeds/samplers.
- The experiments only focus on CIFAR-10 dataset and DDIM samplers. It would be interesting to see how the corrector perform on other datasets/samplers.

**Questions:**

- In the data sampling process from the distribution p_t, why is Gaussian noise added, and how are its mean and variance determined?
- In Algorithm 1, the sampler time horizon T is never used. Should t/h-1 be T/h-1 ?

---

> ### Author Response · Authors · 2025-11-21
> **Resonse to Review**
>
> # Novelty
>
> Thank you for your valuable suggestion. We believe our main contribution is developing a new self-improving methodology for diffusion model based on error analysis, which is a brand-new idea in comparison to generation-verification gap. Using flow-matching is just a way to show our analysis can apply to the real diffusion and brings significant improvement. Our work proves that beyond generation-verification gap, there exists a simple and elegant framework, which can bring the self-improving capability to diffusion models based on error analysis. We also call for the community to work on self-improving methodology beyond generation-verificaion gap framework.
>
> # Writing
>
> Thank you for your valuable comment. We will further clarify the definition of testing and training scalabikity in the introduction section. Thank you for your thoughtful advice.
>
> Also, please let me further elaborate the definition of the testing and training scalability. The **testing scalability** means as we invest more computational budget during the inference time, the performance, which we use FID as metric for the image generation, should grow. The **training scalablity** represents as we invest more computational resource in training, the performance should increase. Therefore, to mesure the scalability of a methodology, we use **performance gain per computational budget**. During inference time, our paper show that **as we invest more computational budget, the FID goes down way more faster than predictor-corrector and DDIM baseline**.
>
> # Corrector Overfitting
>
> Thank you so much for your insightful advice. Our corrector is harder to train than regular flow matching or diffusion models because it requires more sophisticate flow learning. Therefore, we would like propose a simple re-initialization and fine-tuning trick to stablize the training and reduce 95\% training cost. We freeze the pre-trained DDPM by 95\% and fine-tune it with 1e-4 learning rate on 10K synthetic dataset and full training dataset. For each correction timestep, we repeated the experiment by 3 times and report the average FID across 3 runs. We report the final FID scores for correction timesteps on 800, 200, and 2.
>
> | **Corr Timestep** | **FID**              |
> |-------------------|----------------------|
> | 800               | 5.021 ± 0.051        |
> | 200               | 5.133 ± 0.020        |
> | 2                 | 5.186 ± 0.088        |
>
> Also, here is the FID score, training loss, and validation loss during training,  (https://imgur.com/a/lQe8rlw). As we can see, both FID and losses decrease stably during training. Note that the timestep shown in the figure is the complementary of Corr Timestep of **Corr Timestep** shown in the table, which means "corr_trained_timestep: 200" in the figure corresponds to "Corr Timestep 800" in the table, "corr_trained_timestep: 800" represents "Corr Timestep: 200", and "corr_trained_timestep: 998" in figure should be interpreted "Corr Timestep: 2" in table.
>
> With such simple re-initialization and fine-tuning trick, we can achieve similar performance as original paper with only 5\% trainable parameters stably. It prove that our method is a reliable light-weight module to achieve self-improving for diffusion model.
>
> # Generalization for More Advanced Sampler
>
> Here we will show the performance of our method on PNDM. We initialize the corrector from the pre-trained DDPM on CIFAR10, freeze 95% layers of it, and train with 1e-4 learning rate and 50K synthetic dataset. As we can see, our method is generalizable to PNDM and still bring significant quality improvement. Additionaly, we also visualize the FID score and loss during training here (https://imgur.com/a/MfeUuqY), we can see our method can stabily converge as training keep going on.
>
> | **Corr Timestep** | **FID**              |
> |-------------------|----------------------|
> | 800               | 5.078 ± 0.138        |
> | 200               | 5.105 ± 0.058        |
> | 2                 | 5.166 ± 0.089        |
>
>
> # Generalization for Various Dataset Size
>
> ## 50K Dataset Size
>
> Here, we provide FID evaluation for 50K synthetic dataset and full training dataset with 95% frozen rate and 1e-4 learning rate. We report the best FID score and loss and FID score during training at here (https://imgur.com/a/VRLzpGt).  As we can see, we can still achieve stable loss and FID convergence
>
> | **Corr Timestep** | **FID**              |
> |-------------------|----------------------|
> | 800               | 4.974 ± 0.100        |
> | 200               | 5.107 ± 0.076        |
> | 2                 | 5.172 ± 0.057        |
>
> # Large-Scale Dataset
>
> Thank you for your valuable advice. We are still training the corrector on CelebA-HQ due to resource limitation. We will bring the results back to here as soon as possible.

---

> ### Author Response · Authors · 2025-11-27
> **Response to Questions of Review TZ1Z**
>
> # Data Sampling from $p_{t}$
>
> Thank you so much for your question. Our corrector aims to bring the generated marginal distribution $\hat{p}\_{t}$ to ground-truth marginal distribution $p\_{t}$ of the forward process $d \bar{\mathbf{x}}\_{t} = - \bar{\mathbf{x}}\_{t} dt + \sqrt{2} d B\_{t}, t \in [0, T], \bar{\mathbf{x}}\_{0} = p^{*}$. Thefore, we solve the forward process  to sample from ground-truth marginal distribution $p_{t}$, which is generate sample from $\mathcal{N} (e^{−(T −t)} \bar{\mathbf{x}}\_{0}, (1 − e^{−2(T −t))}I).$
>
> # Sampler Time Horizon $T$ is Never Used
>
> Thank you so much for your careful review. We will correct the typo.

---

### Meta-Review · Area_Chair_CzpT · 2026-01-13

**Summary:**

This paper introduces a diffusinon algorithm that replaces generation-verification schemes with a flow-matching corrector.
The idea is natural but several valid concerns were raised.

**Reviewer Concerns:**

Several of the reviewers raised concerns about incremental contributions since the main technique is well known and widely used. The experimental evaluation is also very limited to CIFAR10 and one DDIM sampler.

The authors discuss some additional experiments in the rebuttal but this is a major revision that would require additional new reviewing.

Also all the reviewers complained about poor writing and presentation but this can be addressed by the authors if they want to try again for publication in the next top ML venue.

**Reviewer Scores:**

All the reviewers raised similar concerns that can be addressed but this requires many additional experiments and ablations that go beyond the rebuttal.

---

### Decision · Program_Chairs · 2026-01-26

Reject